# Expanding detection windows for discriminating single nucleotide variants using rationally designed DNA equalizer probes

Guan A. Wang[1,2], Xiaoyu Xie[2], Hayam Mansour[2,3], Fangfang Chen[1,2], Gabriela Matamoros[4,5], Ana L. Sanchez[4,5], Chunhai Fan [6] & Feng Li [1,2✉]

Combining experimental and simulation strategies to facilitate the design and operation of nucleic acid hybridization probes are highly important to both fundamental DNA nano-technology and diverse biological/biomedical applications. Herein, we introduce a DNA equalizer gate (DEG) approach, a class of simulation-guided nucleic acid hybridization probes that drastically expand detection windows for discriminating single nucleotide variants in double-stranded DNA (dsDNA) via the user-definable transformation of the quantitative relationship between the detection signal and target concentrations. A thermodynamic-driven theoretical model was also developed, which quantitatively simulates and predicts the performance of DEG. The effectiveness of DEG for expanding detection windows and improving sequence selectivity was demonstrated both in silico and experimentally. As DEG acts directly on dsDNA, it is readily adaptable to nucleic acid amplification techniques, such as polymerase chain reaction (PCR). The practical usefulness of DEG was demonstrated through the simultaneous detection of infections and the screening of drug-resistance in clinical parasitic worm samples collected from rural areas of Honduras.

[1] Key Laboratory of Green Chemistry & Technology of Ministry of Education, College of Chemistry, Sichuan University, 610064 Chengdu, Sichuan, China. [2] Department of Chemistry, Centre for Biotechnology, Brock University, St. Catharines, ON L2S 3A1, Canada. [3] Department of Cell Biology, National Research Centre, Cairo 12622, Egypt. [4] Department of Health Sciences, Brock University, St. Catharines, ON L2S 3A1, Canada. [5] Microbiology Research Institute, National Autonomous University of Honduras (UNAH), Tegucigalpa, Honduras. [6] School of Chemistry and Chemical Engineering, Shanghai Jiao Tong University, 201240 Shanghai, China. ✉email: fli@brocku.ca

Hybridization of complementary nucleic acid strands through specific and predictable Watson–Crick base pairs plays central roles in genomics research[1,2], medical diagnostics[3–6], and DNA nanotechnology[7–11]. Synthetic nucleic acid hybridization probes and primers have been adopted virtually in all technology platforms to ensure the specific recognition, capture, detection, or assembly of nucleic acid sequences[12–25]. Strategies that combine experimental and simulation approaches to guide the design and operation of nucleic acid hybridization probes are highly effective but remain limited[11,16–19]. Herein, we describe a class of simulation-guided nucleic acid hybridization approach, termed DNA equalizer gate (DEG), which drastically expands detection windows for discriminating single nucleotide variants (SNVs) in double-stranded DNA (dsDNA).

A robust hybridization probe shall be both sensitive and specific. However, discrimination of SNVs is challenged by the intrinsic thermodynamic properties of hybridization reactions, where a trade-off between sensitivity and specificity exists[16]. Therefore, extensive experimental and simulation studies have been focused on obtaining an optimal trade-off between hybridization yield (sensitivity) and sequence selectively (specificity) through the design of various frustrated hybridization probes such as molecular beacons[26–28] and toehold-exchange probes[11,16,29], or through tuning experimental conditions such as temperature and denaturing reagents[30,31]. However, with current approaches, detection signals increase monotonically with increases in concentrations of both correct and spurious targets. For any observed detection signal, it may correspond to a low concentration of a correct target but could also be a result of a high concentration of a spurious target. Therefore, a detection window exists between the target concentrations that lead to the same level of detection signals (Fig. 1a). Current frustrating hybridization probes pursue enlarged detection windows by increasing energy barriers for generating detection

signals (Fig. 1b). For example, longer stem domains and allosteric inhibitors have been introduced to molecular beacons to expand their detection windows[32–35]. Enlarged detection windows are also achievable in toehold-exchange probes by elongating the reverse toehold (simulation results in Fig. S1). However, the success of current strategies is at the cost of shifting detection windows towards the higher concentration end, inevitably sacrificing sensitivities at lower concentration range. Different from existing approaches, DEG offers a paradigm to expand the detection window without compromising sensitivity at lower concentration range by transforming the quantitative relationship between the detection signal and target concentrations from a monotonic sigmoidal function to an asymmetric unimodal function (Fig. 1c).

DEG is a DNA computing module that acts on dsDNA and allows user-definable transformation of the quantitative relationship between detection signals and target concentrations (Fig. 2a). Through the transformation, DEG drastically expands detection windows for discriminating SNVs in dsDNA to as much as infinite. DEG also possesses an intrinsic self-competing mechanism that further improves the sequence selectivity. A thermodynamic-driven mathematical model was constructed, where the detection window, reaction yield, and sequence specificity can be precisely simulated and predicted in silico. The practical usefulness of DEG was established through the integration of polymerase chain reaction (PCR) for the simultaneous infection detection and drug-resistance screen in clinical parasitic worm samples collected from school-age children residing in endemic rural areas of Honduras.

## Results

**Design principle**. The goal of designing DEG is to maximize detection windows for discriminating SNVs by suppressing the detection signals for spurious targets through the transformation of the quantitative relationship between detection signals and target concentrations. To quantitatively describe the detection window, we introduce a Robustness Factor (RF) that is defined as the ratio of concentrations between a spurious and a correct target generating the same level of detection signal, $RF = [T]_{spurious}/[T]_{correct}$ (Fig. 2b). As such, the greater the RF value, the wider the detection window. Although DEG acts on dsDNA, the detection of single-stranded DNA (ssDNA) can also be considered as a special case of DEG, where the concentrations of DNA Equalizer Probes (DEPs) approach infinite.

The workflow and principle of DEG are illustrated in Fig. 2. A double-stranded input AB produces a single-stranded target A and its complementary sequence B through a rapid heating at 95 °C and then snap cooling to 0 °C in a thermal protocol. B is then consumed by DEPs that are of the identical sequences with A forming three-stranded complex BCD (Fig. 2b). The yield (η) of A is thus determined quantitatively by the concentration of DEPs. When the concentration of AB is less than those of DEPs, A is the predominant product, although a competition exists between A and DEPs for hybridizing to B (Fig. 2c). When the concentration of AB is greater than those of DEPs, unconsumed B will rehybridize with A in the renaturation process (Fig. 2d). Therefore, a maximum yield exists when the concentration of AB equals to those of DEPs. Finally, net A is quantified using a toehold-exchange reporter which is designed to be sensitive to SNVs. As each DEP is designed to only contain the sequence of either the toehold domain or the branch migration domain of the reporter, no fluorescence signal can be produced in the absence of the target (Fig. S15). Through DEG, a conventional sigmoid detection curve of hybridization probes is transformed into an asymmetric unimodal one (Fig. 2d).

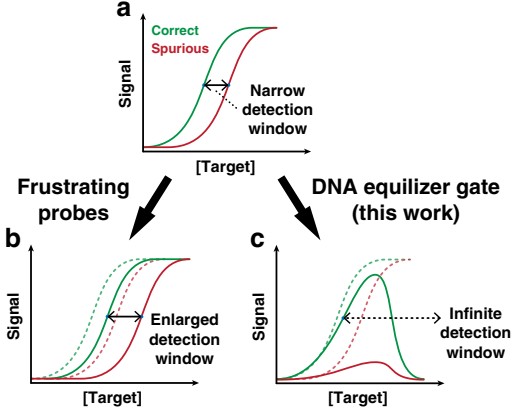

**Fig. 1 Detection windows for discriminating single nucleotide variants. a** Typical titration curves for analyzing correct and spurious targets using frustrating nucleic acid probes such as toehold exchange or molecular beacon. Detection signals for both correct and spurious targets increase monotonically with increases in target concentrations. As such, a detection window exists between target concentrations that lead to the same detection signal. **b** Enlarged detection windows may be achieved by shifting the trade-off between sensitivity and specificity through a stronger reverse toehold in a toehold-exchange beacon or a longer stem domain in a molecular beacon. **c** DEG expands the detection window by physically transforming the quantitative relationship between the detection signal and target concentrations through a DNA computing module. As a result, detection signals for a spurious target are suppressed throughout the concentration ranges, which ensures the high specificity and robustness for discriminating SNVs.

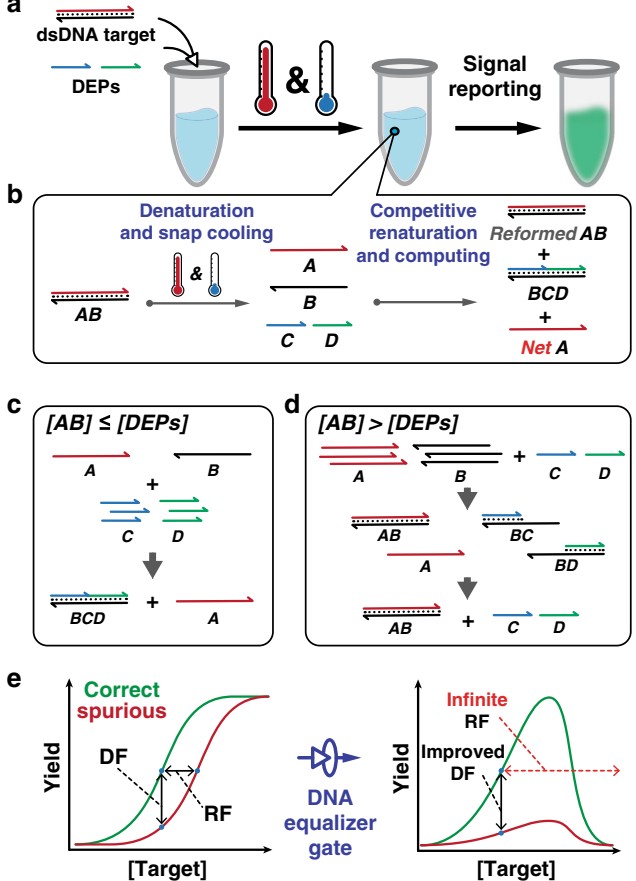

**Fig. 2 Schematic illustration of the DNA Equalizer Gate (DEG). a** The overall workflow for quantifying dsDNA using DEG. A mixture of target dsDNA and DNA Equalizer Probes (DEPs) is heated and rapidly cooled to produce ssDNA outputs with well-controlled quantity using an autonomous molecular computation in the test tube. Fluorescence signals are then generated via a reporter probe. **b** Mechanistically, the dsDNA target (AB) is denatured into A and B during a heating and snap cooling procedure. A competition between DEPs (C and D) and A then occurs for hybridizing with B during renaturation. The net amount of the ssDNA output (A) is quantitatively determined by an autonomous computing process that compares the initial concentrations between the target and DEPs. **c** When [AB] ≤ [DEPs], the reaction between B and DEPs (i.e. the formation of BCD) is thermodynamically favored, which maximizes the production of A. **d** When [AB] > [DEPs], BC and BD are generated as intermediates, which then consumes A through strand displacement. **e** Through this computing process, DEG transforms the quantitative relationship between the detection signal and target concentrations from a typical sigmoidal function to a unimodal function. As such, detection signals for a spurious target is significantly suppressed, enabling a much-enlarged detection window and improved discrimination factor (DF).

Comparing to existing strategies via manipulation of energy barriers of frustrating probs, transformation of the quantitative relationship between detection signals and target concentrations from a sigmodal function to an assymetric unimodal one offers three distinct advantages. First, the transformation only suppresses detection signals at higher concentration end. As such, it allows the dramatic expansion of detection windows without compromising the sensitivity at the lower concentration end (Fig. S6). Second, the manipulation of detection window is user-definable and can be achieved at any target concentration. In principle, correct and spurious targets achieve maximum yields simultaneously in DEG, both of which are governed by the

concentration of DEPs. As such, a detection window is definable and tunable by simply altering the concentration of DEPs. Moreover, the detection signal for a correct target remains to be much higher than that of a spurious one throughout concentration ranges (RF = ∞, Fig. 2e right), whereas the detection window for a conventional probe is much narrower (Fig. 2e. left, Fig. S6). Third, as detection signals for the spurious target are significantly suppressed, discrimination factor (DF) is significantly enhanced through a wide concentration range (Fig. 2e right). At molecular level, B serves as a molecular sink that competitively consumes A regardless the identity or the position of the mutation, which is significantly different from existing strategies harnessing molecular sinks or reservoirs[36,37] that are designed specifically for known mutations. To quantitatively simulate and predict the effectiveness of DEG for expanding the detection window and for improving sequence selectivity, a theoretical model was established and detailed in the next section.

**Theoretical model**. Here, a mathematical model was introduced to quantitatively profile DEG by taking all possible reactions into consideration (Fig. 3a). To derive the yield of each DNA species in this reaction network as a function of both sequence design ($\Delta\Delta G^0$) and equalizer probe concentrations, a set of eight equilibrium equations need to be solved. However, we found that these equations were coupled to one another, which was mathematically difficult to solve. Therefore, a stoichiometric matrix RM was introduced to help simplify the calculation (Fig. 3a), where the first four rows were ranked to be essential (details in Supplementary Information section S2.4). This essential set of equilibrium equations was then solved by a numerical approach, where distributions of A and AB were solved as a function of the target concentration and plotted in Fig. 3c.

The thermodynamic-driven model successfully predicted the distribution of A and AB at the concentration range, where [AB] > [DEPs] (Fig. 3c). However, it failed to simulate the thermodynamic behavior of DEG when [AB] ≤ [DEPs]. We found that a probability function that took the possible distributions of DEPs on B was necessary to correctly reflect the final equilibrium distribution of each DNA species (Fig. 3d and Fig. S5). Mathematically, the probability for the successful formation of a BCD complex is $([DEPs]_0/[AB]_0)^2$ (Fig. 3d). The combination of the thermodynamic-driven model with probability correction leads to a characteristic asymmetric unimodal curve (Fig. 3e), which was also confirmed experimentally (details in the next section).

**In silico prediction and experimental validation**. Using our theoretical model, η, DF, and RF were firstly quantitatively profiled in silico against three critical factors in DEG, including the target concentration, the sequence design ($\Delta\Delta G^0$), and the detection window defined by DEPs. The detection of ssDNA may also be described in our model by setting the concentration of DEPs to be infinite, where the yield for producing A is 100%. Simulation results in Fig. 4 depict the theoretical transitions from the detection of ssDNA ([DEPs] = ∞) to the detection dsDNA with varying concentrations of DEPs at 50, 100, 200, and 500 nM. Unlike conventional frustrating probes (toehold exchange or molecular beacon) where η is saturated beyond a certain target concentration (Fig. 4a), a maximum η exists in DEG at a single target concentration that is defined exclusively by DEP ([T]_max = [DEPs]) and is sequence-independent (Fig. 4b). The simulation results also reveal a significant expansion of the detection window where highly specific discrimination of single nucleotide mutations can be achieved (Fig. 4d). The level of improved DF is also definable by the concentration of DEP (Fig. 4d). As η for high

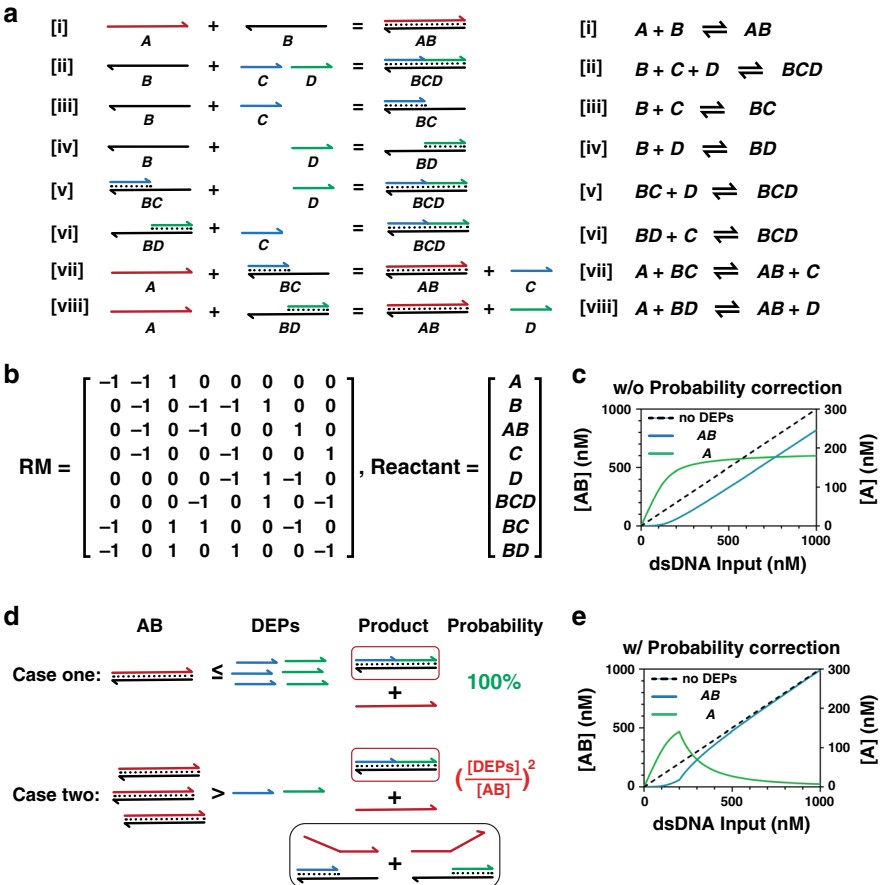

**Fig. 3 Theoretical model of DNA Equalizer Gate (DEG). a** Schematic illustration of all possible elemental reactions occurring in the DEG. **b** Linearization of the complex reaction network in DEG into **0 = RM·Reactant** to extract the independent equations, where RM is the stoichiometric matrix and Reactant represents DNA species. The rank of RM was determined to be 4, indicating that four independent equilibrium equations need to be solved. As such, reactions [i - iv] were chosen to build mathematical model. **c** In silico prediction of the yields of A and AB as a function of the concentration of dsDNA target without performing the probability correction. **d** Schematic illustration of the need for probability correction when [AB] > [DEPs]. As each DEP binding is an independent event, the multiplicity rule was applied here. When [AB] ≤ [DEPs], the probability that two DEPs bind to the same B is 100%. However, when [AB] > [DEPs], the probability becomes dependent on the ratio between the initial concentrations of AB and DEPs, where Probability = $([DEPs]_0/[AB]_0)^2$. **e** In silico prediction of the yields of A and AB as a function of the concentration of dsDNA target with the probability correction.

concentrations of SNVs has been suppressed exclusively, a remarkable transition of RF is observed from finite values (Fig. 4e) to infinite (Fig. 4f).

The experimentally measured η, DF, and RF at varying concentrations of a synthetic dsDNA target are plotted in Fig. 5 for comparison with those predicted in silico. Experimental validation and optimization are detailed in Supplementary materials section S3 (Figs. S10-S16). A correction of $\Delta G_{rxn}^0$ by +1.58 kcal/mol was found to significantly improve the agreement between experimental observation and in silico prediction (Fig. S7). η and DF at a specific target concentration were calculated directly using fluorescence readout from the reporter. Consistent with in silico prediction, maximum η were observed for both correct and spurious targets, which was defined strictly by the concentration of DEP (Fig. 5a). As theoretically predicted, η for the spurious target is significantly suppressed by DEG. As a result, improved DF was also observed, which also agreed well with simulation (Fig. 5b). RF was measured indirectly by first fitting a calibration curve using a non-linear model and then calculated according to the definition (S2.2, eq. S8, Fig. S8). Again, infinite RF was determined across wide concentration ranges (Fig. 5c). The effectiveness and flexibility of DEG were further verified experimentally against varying types and

locations of single nucleotide mutations (Fig. 5d, e), varying length of dsDNA targets (Fig. S17–S19), and finally nine sets of clinically important SNVs (Fig. S31-S33). DEG works well for all sets of targets except when mutation occurs at the very edge of the dsDNA (Fig. 5e).

**Integration of DEG with PCR.** A practically applicable DNA hybridization probe shall compatible with commonly used nucleic acid amplification techniques, such as PCR. As DEG acts directly on dsDNA, it is an ideal probe for analyzing dsDNA amplicons. Therefore, we next verified the adaptivity of DEG to PCR. As a proof-of-principle, a set of four DEPs were designed for a representative 87 bp dsDNA amplicon (Fig. 6a), which was shown to be fully compatible with DEG (Fig. 6b, S37 and S38). To avoid potential cross-reactions, two outer DEPs were designed intentionally to be identical as the PCR primers (Fig. 6c and Fig. S36).

Results in Fig. 6d demonstrate that the DEG-PCR is both highly sensitive and specific. As low as 1 aM synthetic DNA templates were detectable. More importantly, fluorescence signal for 1 pM spurious template containing a single nucleotide mutation is significantly suppressed using DEG, which is much lower than that of 10 aM of the correct template (Fig. 6d). By

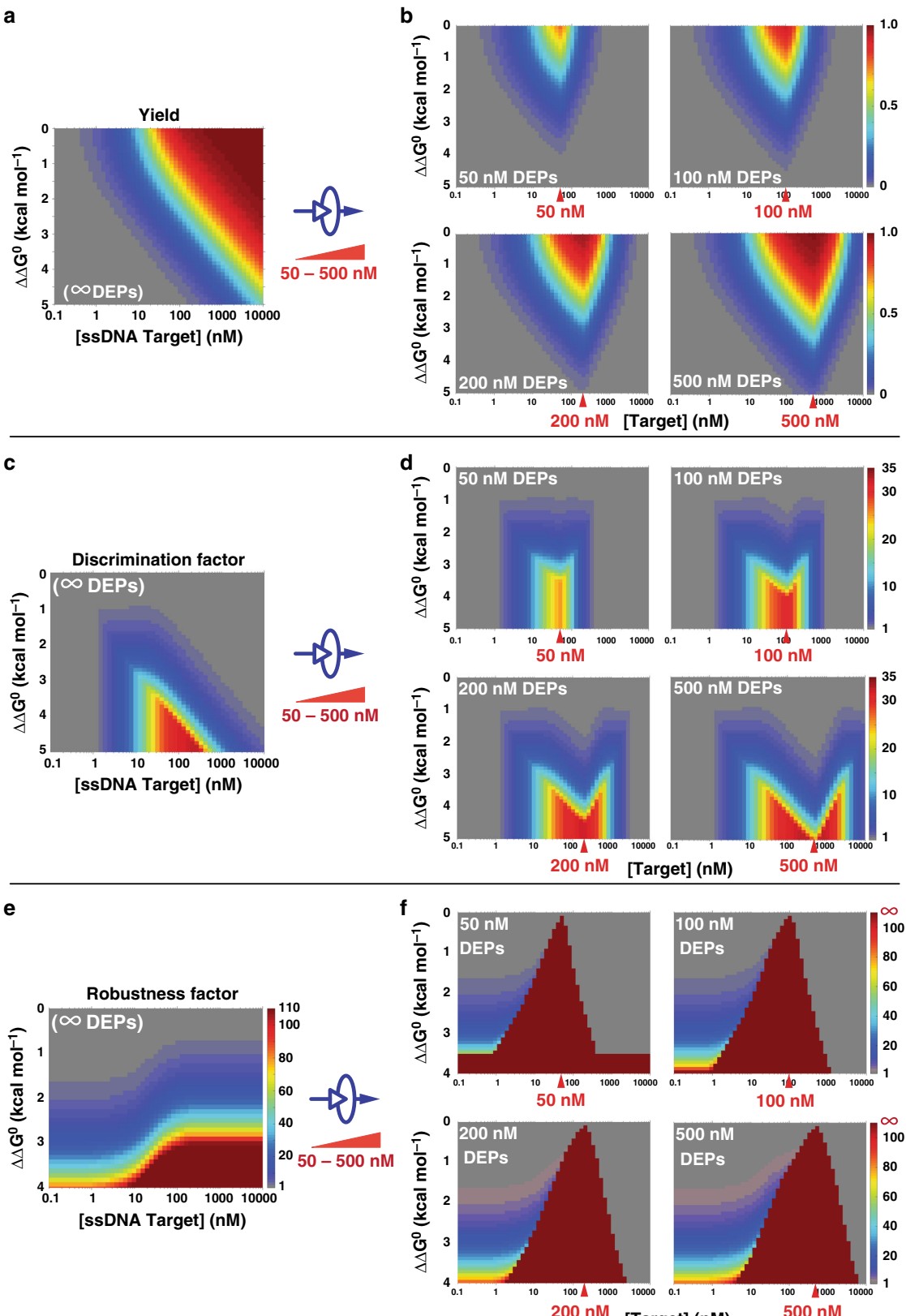

contrast, a much narrower detection window (above 1 fM) was observed when asymmetric PCR was used to generate detectable ssDNA amplicon followed by the readout using the same toehold-exchange reporter (Figs. 6e, f and S39).

**Clinical validation of DEG-PCR.** We finally employed DEG-PCR for the diagnosis of soil-transmitted helminth (STH) infections with clinical samples collected from school-age children living in highly endemic rural areas in Honduras. STH infections

**Fig. 4 Simulation results of DNA Equalizer Gate (DEG).** In silico prediction of the reaction yield as a function of both target concentration and $\Delta\Delta G^0$ for classic toehold-exchange (**a**) and DEG of varying DEP concentrations at 50, 100, 200, and 500 nM (**b**). The classic toehold-exchange can be considered as a special case of DEG, where [DEPs] = ∞. Maximum yields exist for DEG, where [AB] = [DEPs]. Yields of spurious targets are significantly suppressed across wide concentration ranges, which can help improve the specificity and expand the detection window. In silico prediction of discrimination factors for classic toehold-exchange (**c**) and DEGs (**d**). The detection window for discriminating SNVs is tunable by altering the concentrations of DEPs. In silico prediction of robustness factors for classic toehold-exchange (**e**) and DEGs (**f**). The use of DEG dramatically increases RF values from a finite value to infinite.

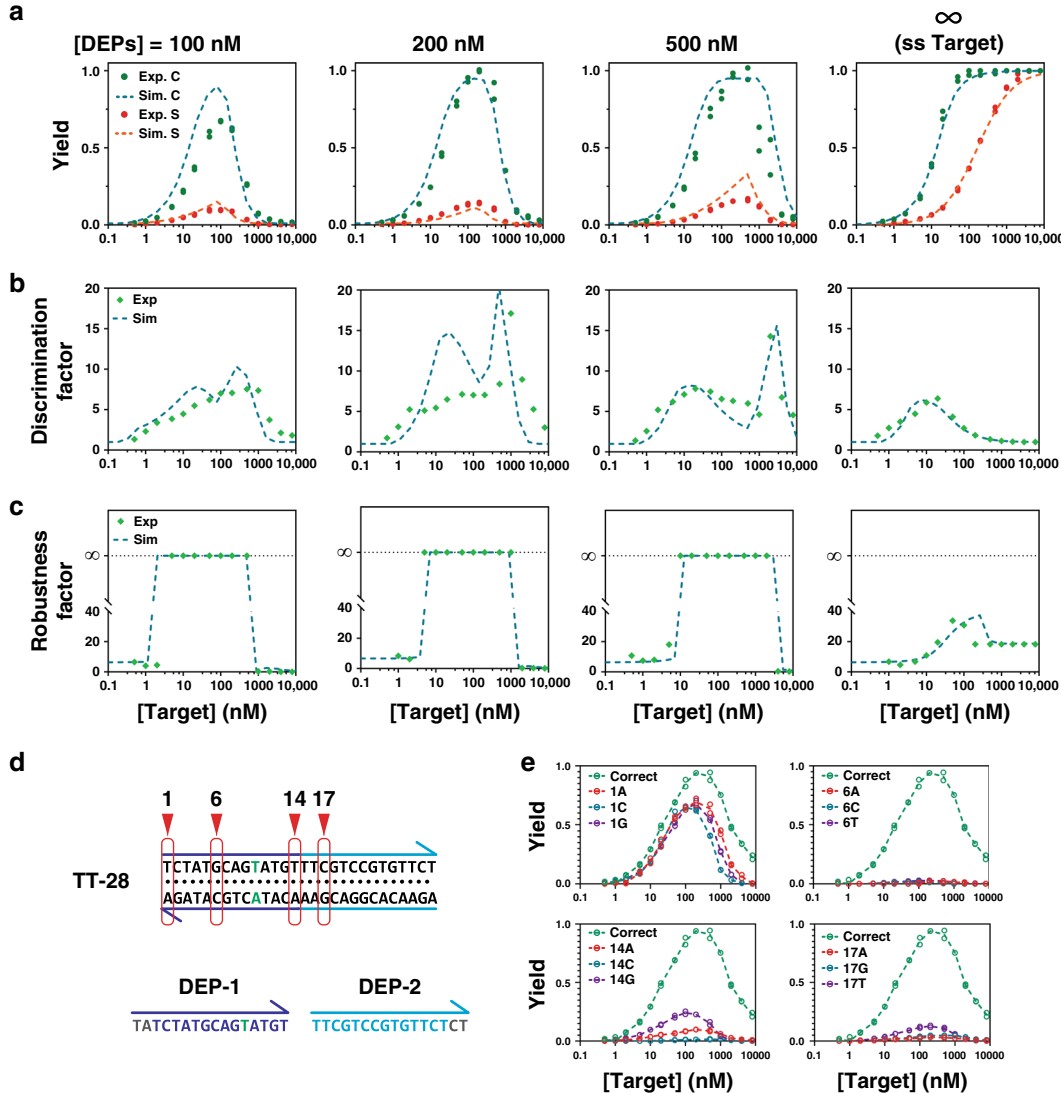

**Fig. 5 Experimental validation of DNA Equalizer Gate (DEG).** **a** Experimentally determined yields (Exp) plotted against target concentrations for DEGs with varying DEP concentrations and compared to simulation (Sim). The classic toehold-exchange can be considered as a special case of DEG, where [DEPs] = ∞. Individual replicates (n = 2) are shown as dots. **b** Experimentally determined discrimination factors using a pair of synthetic correct and spurious targets ($\Delta\Delta G^0 = 2.29$ kcal/mol) plotted against target concentrations and compared to those predicted in silico. **c** Robustness factor plotted against target concentrations and compared to in silico prediction. **d** Schematic illustration of sequences of the target and DEPs. Single-nucleotide mutations were made to the target at positions 1, 6, 14, and 17. **e** Experimentally determined yields plotted against target concentrations for correct and spurious targets carrying mutations at four designated positions. Individual replicates (n = 2) are shown as circles. All experiments were run at 37 °C in 1 × PBS buffer with 1 mM $Mg^{2+}$ and 20 nM of toehold-exchange beacons. Source data are available in the Source Data file.

are global health issue, affecting more than 1.5 billion world's population[38]. The extensive drug usage (e.g., Albendazole) for treating STH infections in endemic countries or regions has created issues of drug resistance[39,40]. As such, an ideal diagnostic test for STH infection shall allow simultaneous detection of STH infection and screen for drug resistance (D.R.).

Thus motivated, we employed DEG-PCR for the detection of STH infections, meanwhile screening for drug resistance in the same assay (Fig. 7a). Two fluorescence reporters were designed to test codon 196 to 203 and codon 206 to 213 of the β-tubulin gene of *Trichuris trichiura* (Fig. 7b). A single-nucleotide A to T mutation at the 200th codon of β-tubulin is a well-established

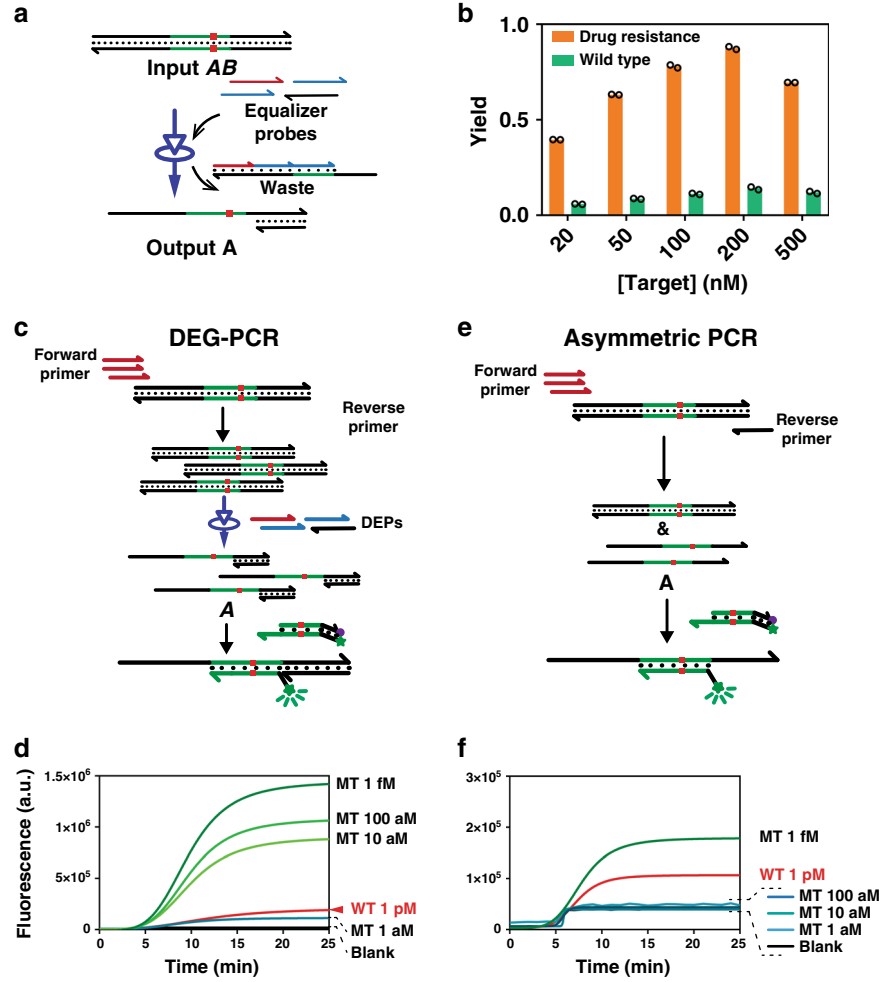

**Fig. 6 Integration of DNA Equalizer Gate (DEG) with polymerase chain reaction (PCR). a** Schematic illustration of analyzing dsDNA using a 4-DEP design. **b** Experimental validation of the 4-DEP design for the detection of an 87 bp dsDNA as a mimic of PCR amplicon. The concentration of outer DEPs was fixed at 500 nM and that of inner DEPs was set to be 200 nM. Individual replicates ($n = 2$) are shown as circles. **c** Schematic illustration of DEG-PCR using a 4-DEP design. The outer two DEPs (red and black) are designed to be identical with PCR primers. **d** Real-time monitoring of DEG-PCR using a toehold-exchange reporter. A wide detection window was achieved, where as low as 10 aM of correct template could be clearly discriminated from 1 pM spurious target containing a single nucleotide mutation. **e** Schematic illustration of the asymmetric PCR followed by the detection using a toehold-exchange reporter. **f** Real-time monitoring the detection of asymmetric PCR amplicon revealing a much narrower detection window than DEG-PCR, where correct discrimination could be made only above 1 fM. Source data are available in the Source Data file.

genetic variant for drug-resistance screening (Fig. S40)[40]. The toehold-exchange reporter testing this domain (codon 196 to 203) was thus designed to be highly sensitive to this SNV by including a 5-nt reverse toehold, whereas no reverse toehold was designed for the reporter targeting codon 206 to 213. The two reporters were labeled with spectrally distinct fluorescent dye (FAM and Cy5) and thus operated simultaneously in solution (Figs. S41 and S42). Synthetic DNA standards of varying concentrations and 13 clinical TT samples with negative resistance were first tested using the dual-channel DEG-PCR (Figs. S43-46) and plotted in Fig. 7c, where three regions (error eclipse with 99% confidence) were definable representing positive infection and positive resistance (D.R.+), positive infection but negative resistance (D.R.−), and no detectable infection (N.C.). Six clinical parasitic specimens expelled by patients who received Albendazole treatment in Honduras were tested and found to be TT positive but no drug resistance (Fig. 7d). Two clinical Ascaris worm specimens serving as negative control were also tested and found to be TT negative. All results were consistent with diagnostic tests using microscopy (Kato-Katz), post-PCR gel analysis (Fig. S47) and DNA sequencing (Fig. S48).

## Discussion

We have introduced DEG, a class of nucleic acid hybridization probes for the direct analysis of dsDNA with the user-definable expansion of detection windows and improved sequence selectivity. Using DEG, the quantitative relationship between the detection signal and target concentrations was transformed from a sigmoidal function to an asymmetric unimodal one, where maximum yield exists at a single target concentration that is defined exclusively by DEP ($[AB]_{max} = [DEPs]$) and is sequence-independent. Therefore, unlike conventional hybridization probes where the detection signal of a spurious target will eventually catch up that of a correct one[33], signals for spurious targets remain to be suppressed in DEG despite the increases in target concentrations and RF will eventually become infinite. Because of the mathematical transformation, the same detection signal may correspond to two concentrations of the same correct target (Fig. 8). This will not cause any issue for discriminating single nucleotide mutations, as the detection signal remains to be higher than any concentration of the mutated targets. For further quantifying the correct target using DEG, we found that the inclusion of a

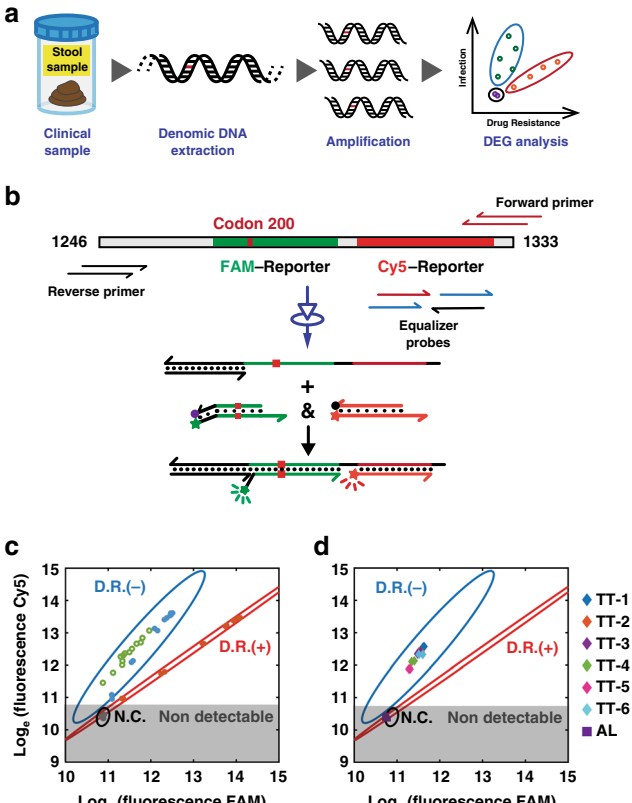

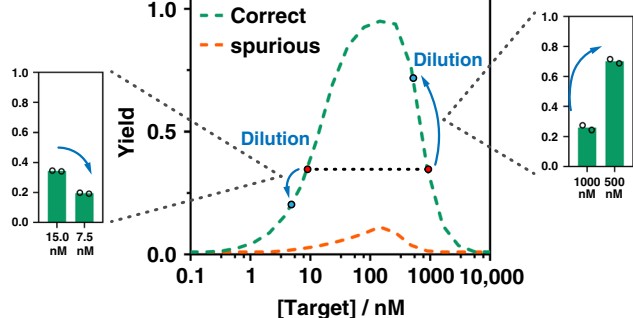

**Fig. 8 A dilution strategy to narrow the concentration range for quantifying target nucleic acid.** Experimental demonstration of a representative scenario, where the same detection signal corresponds to two possible concentrations of the same target. The yield of 0.35 corresponds to either 15 nM or 1 μM of the correct dsDNA target. After a 2 × dilution step, the original 1 μM target concentration (500 nM after dilution) produced an even higher detection signal (0.8) whereas the signal of the original 15 nM target concentration (7.5 nM after dilution) decreased to 0.2. Individual replicates ($n = 2$) are shown as circles. Source data are available in the Source Data file.

**Fig. 7 Application of DEG-PCR to analyzing clinical parasitic worm samples. a** A typical workflow for analyzing parasitic worm (*Trichuris trichiura*, TT) specimens collected from stool samples of school-age children in the rural areas of Honduras followed by the detection using DEG-PCR. **b** Simultaneous detection of parasitic infection and screening for drug-resistance was achieved using a dual-channel design (FAM- and Cy5-Reporter). PCR primers were designed to amplify nucleotide 1246-1333 in the β-tubulin gene, containing the 200th codon. A single nucleotide A to T mutation of this codon is a hotspot for drug resistance screening. A toehold-exchange reporter (FAM-reporter, green duplex) labeled with FAM was used to discriminate this point mutation, whereas a strand-displacement reporter with no reverse toehold (Cy5-reporter, red duplex) was employed to detect a conservative region near codon 200. Experimental tests of the dual-channel DEG-PCR using synthetic DNA standards (blue and red dots) and 13 (D.R.−) clinical samples (green circles) as a training set (**c**) and 8 unknown clinical parasitic worm samples (**d**). Test results are classified into three areas defining the positive infection and drug resistance (D.R.+), positive infection and no drug resistance (D.R.−), and negative infection (N.C.). Error eclipses with 99% confidence interval and 2-degrees of freedom (two fluorescence channels) were used to define D.R.+ and D.R.−. Eight clinical worm specimens including six *Trichuris trichiura* worms (TT-1 to TT-6) and two *A. lumbricoides* worms (AL, as negative controls) were tested and plotted in (**d**). Source data are available in the Source Data file.

dilution step would effectively narrow the target concentration (Fig. 8).

The ability to transform quantitative relationship between the detection signal and target concentrations sets the DEG approach apart from existing nucleic acid hybridization probes. Prior to our study, the concentration dependency of DNA hybridization probes has been emphasized and tested. For example, Zhang and colleagues have introduced an R value that was experimentally defined by the horizontal distance between the calibration curves of a correct and that of a spurious target at 50% yield[17]. Ricci and

colleagues have experimentally examined and tuned useful dynamic ranges of molecular beacons in response to varying allosteric designs[33–35]. However, existing solutions focus on manipulating reaction energy barriers rather than altering the monotonic quantitative relationship between detection signals and target concentrations.

Another advantage of DEG is the fact that it acts on dsDNA using a simple, one-step, wash-free, and enzyme-free procedure but produces ssDNA output in a highly quantitative and predictable manner. This differs significantly with existing strategies where enzymes and tedious procedures were often used to generate toehold domains for dsDNA[17,41] or to remove antisense strands to produce ssDNA[42]. The unique design of DEPs makes them fully compatible with upstream nucleic acid amplification techniques such as PCR, and downstream detection probes, such as toehold-exchange beacons, with minimal signal leakages (Fig. S15), which eliminates the need for the removal of antisense ssDNA through enzymatic degradation or denaturing followed by separation. As such, the principle of DEG can be employed to develop diverse assays for point-of-care applications.

More importantly, instead of treating the antisense ssDNA as a waste[30], our DEG system takes it as a molecular sink that competitively consumes the sense ssDNA once the mutation exists. Unlike existing molecular sink or reservoir created to enhance sequence selectivity[36,37], the design of which requires prior knowledge of the specified mutation, the DEG splits a dsDNA and thus produces a corresponding sink for any mutation. We demonstrated that DEG approach is particularly effective for discriminating challenging mutations, such as a single nucleotide A to G mutation (because of the formation of G-T wobble), where both DF and RF have been drastically improved within a wide detection window (Figs. S28–S30).

Collectively, our DEG approach demonstrates remarkable analytical performance for analyzing mutations in genetic markers (Figs. S32 and S33). When comparing to asymmetric PCR, a widely used strategy to produce ssDNA amplicons or to prepare double-stranded toehold-exchange probes, our DEG-PCR shows significantly better sensitivity, improved specificity and wider detection window. The practical usefulness of DEG has also been successfully verified using clinical STH parasitic worm samples collected at endemic regions. The capability of simultaneous detection of parasitic infection and drug-resistance screening will

make our strategy an idea tool for genetic analysis in diverse clinical settings.

Notably, the analytical performance of our DEG approach has not only been demonstrated experimentally, but also been quantitatively and precisely predicted through simulation. Our success in combining thermodynamic parameters calculated using NUPACK software and numerical approaches using MATLAB further echoes the programmable and predictable nature of the Watson-Crick base pairing rule[43,44]. The ability to make accurate mathematical predictions for systems involving complexed reaction pathways, such as the DEG system in this work, reveals again the power of in silico tools to help guide the rational design of nucleic acid hybridization probes and DNA-mediated biosensors. Therefore, we anticipate our effort in developing DEG approach may benefit both fundamental research in DNA nanotechnology and practical uses of nucleic acid hybridization probes to real-world applications.

## Methods

**DNA oligonucleotides**. The DNA oligonucleotides used in this study were purchased from Integrated DNA Technologies (IDT, Coralville, IA). Fluorophore (FAM- and Cy5-) and quencher (Iowa Blank) modified DNA oligonucleotides were purified by IDT using high-performance liquid chromatography (HPLC). Other DNA species were used as purchased without further purification. Sequences and modifications of all oligonucleotides are listed in Table S1.

**Buffer conditions**. DNA oligonucleotides were re-suspended by dissolving oligonucleotides using $1 \times$ tris-EDTA (TE) buffer (10 mM Tris-HCl, pH 8.0, 1 M EDTA, purchased from Sigma as $100 \times$ stock) and then stored at $-20\,°C$. Unless indicated otherwise, $1 \times$ TE buffer containing 10 mM MgCl$_2$ and 0.5% (v/v) TWEEN 20 (Sigma) was used as the molecular reporter buffer. $1 \times$ PBS (pH 7.4, purchased as $10 \times$ PBS stock from Sigma) buffer containing 1 mM MgCl$_2$ and 0.5% (v/v) TWEEN 20 was used as the reaction buffer. TWEEN 20 was used to prevent the potential loss of DNA oligonucleotides during dilution and pipetting.

**Preparation of fluorescent reporters**. All strand-displacement (SDR) and toehold-exchange (TER) reporters were annealed using a BioRad T100 thermo-cycler in molecular reporter buffer. The samples (typically at a final concentration of 5 μM) were heated to 95 °C for 5 min, and subsequent gradually cooled to room temperature at a constant rate over a period of 40 min. Particularly, the quencher to fluorophore concentration ratio used for SDR was 1.5 and that of TER was 3. Prepared reporter solutions were stored in 4 °C until use.

**Mathematical model building**. Free energy of DNA strands and complexes were estimated by NUPACK. For thermodynamic parameters setting of DEG, the temperature was set to 4 °C (in ice-water bath), concentration of Na$^+$ was 0.1 M, and Mg$^{2+}$ was 0.001 M; whereas temperature setting for DNA species in toehold-exchange reaction was 37 °C. Other parameters were used set as default setting.

Analytical solutions of concentration-dependence equations for η, DF, and RF were calculated through symbolic approach in MATLAB (2019a, MathWorks). Matrix (RM) analysis and solving equilibrium equations system were performed in the same platform. Particularly, numerical computing approach is necessary due to highly coupled variables in the equation system (3$^{rd}$ order). Boundary conditions were restricted to real value and reasonable answers (for example, yield needs to be larger than 0 yet smaller than 1). To calculate theoretical RF values, two reverse functions were adopted: first one was to convert yield (normalized) to the concentration of ssDNA target; second one was to reverse the ssDNA target back to corresponding concentration of dsDNA with the consideration of probability function. Experimental RF values were calculated by fitted non-linear curve functions (see details in Supplemental materials section S2.6.2). Two-dimensional curves were plotted in GraphPad Prism 8.0.1, and three-dimensional heat maps were drawn in MATLAB.

**Protocols for using DEG**. An ice-water cooling bath was prepared (4 °C) in advance. The dsDNA target and DEPs with user-defined concentration were mixed in a 0.2 mL PCR tube, adjusting volume to 100 μL. Sample tubes were then placed in a thermal cycler (Bio-Rad T100$^{TM}$) and heated up to 95 °C for 5 min (setting as 10 min with redundancy for next step). While samples were kept hot in thermal cycler, tubes were rapidly transferred and immersed inside ice-water bath for 2 min (Fig. S9). 90 μL of the samples were translocated to microplate (Corning) and warmed for 5 min within a microplate reader (Molecular Devices) which was set as 37 °C. Thereafter, 10 μL of 200 nM toehold-exchange reporter was added to trigger the reaction.

**DEG-PCR**. In a typical PCR protocol, 4 μL of DNA template, 20 μL Taq $2 \times$ Master Mix, and proper concentration (typically 500 nM) of forward and reverse primers (Table. S1. Forward and Reverse primer) were mixed to a final volume of 40 μL. PCR was initiated by incubation at 94 °C for 3 min then followed by 35 cycles (denaturation at 94 °C, annealing at 52 °C, and extension at 72 °C for 30 s each) and a final extension at 72 °C for 5 min in a Bio-Rad T100$^{TM}$ thermal cycler. The thermal protocol of asymmetric PCR remained the same, whereas with unbalanced primer concentration (500 nM forward primer and 40 nM revere primer, Fig. S30). The PCR amplicons were then mixed with 4 DEPs and adjusted volume to 90 μL. To avoid potential side reactions, the outer DEPs (identical with primers) were set to 500 nM and inner two DEPs were 200 nM. A typical DEG protocol was followed, and dual reporters (separate FAM and Cy5 fluorescent channels) were added to embark reaction.

**Time-based fluorescence studies**. SpectraMax i3 microplate reader (Molecular Devices) was used to acquire real-time fluorescence data and analyzed in Excel 2016. Temperature was set to 37 °C and fluorescence was monitored in a frequency of 1 data point per minute for 1 h. The excitation/emission wavelength for FAM channel was set 485 nm/515 nm and that of Cy5 channel was 640 nm/675 nm. Fluorescence data were normalized and converted into apparent hybridization yield according to formula $\eta = (F - F_b)/(F_m - F_b)$, where $F$ is the sample fluorescence readout at equilibrium, $F_m$ denotes the maximum fluorescence observed for 50-fold excess of correct ssDNA target to strand-displacement beacon, and $F_b$ is the background fluorescence generated by protected beacon only. For practical purpose, equilibrium fluorescence data were collected at around 20~30 min when reaction roughly reached to equilibrium.

**Analyzing STH clinical samples using DEG-PCR**. STH worm specimens were recovered from eight school-age children in the rural region of La Hicaca located in the northwestern area of Honduras[45]. All STH-infected children were treated by the health center's registered nurse as per national guidelines. A subgroup of eight children harboring infections of heavy and moderate intensity were invited to receive a special deworming treatment with the aim of recovering adult parasite specimens. The treatment schedule was administered by the health center's nurse as described previously[45]. The eight participants received a treatment scheme based on pyrantel pamoate and oxantel pamoate (Conmetel) during the first three days and Albendazole during a fourth day. The adult worms expelled in feces were washed with saline solution and stored in 70% ethanol. Following the recovery of specimens, DNA was extracted using the Automate Express DNA Extraction System (Thermo Fisher Scientific Inc.) with the commercial kit PrepFiler Express BTA, according to the manufacturer's protocol. Thereafter, a typical DEG-PCR procedure (250 nM of each PCR primer; 200 nM of each DEP) was followed to detect these clinical DNA samples.

Two batches (duplicates in each batch) of synthetic DNA templates from 1 aM to 1 pM (containing D.R.(−) and D.R.(+)) using the same DEG-PCR protocol as well as 13 clinical parasitic worm samples to build the fluorescence distribution map. Error eclipses with 99% confidence interval and 2-degrees of freedom (two fluorescence channels) were used to define D.R.(+) and D.R.(−).

**Ethics statement**. This study received clearance from the Bioscience Research Ethics Board of the Brock University (file number 13-195), as well as from the Research Ethics Board of the master's Program in Infectious and Zoonotic Diseases (MEIZ), School of Microbiology, National Autonomous University of Honduras (UNAH). Informed consent was obtained from the parents or legal guardians and was documented with research participants' signatures or fingerprints on the consent forms that had been fully explained to them. Upon parents or guardians' consent, children were invited to enroll in the study and those willing to participate provided verbal assents that were documented on a child assent form with the signature of a third-party witness.

**Polyacrylamide gel electrophoresis**. A 5 μL PCR amplicon solution was mixed with loading buffer (Bio-Rad) and then loaded on 8% native PAGE gel to verify and estimate the PCR procedure. A voltage of 110 V was used to drive the electrophoresis. Thereafter, the gel was stained with Ethidium Bromide and imaged using Gel Doc XR + Imager System (Bio-Rad).

**Reporting summary**. Further information on research design is available in the Nature Research Reporting Summary linked to this article.

## Data availability

All other relevant data are available upon reasonable request. Source data are provided with this paper.

## Code availability

The Matlab code for all simulated results, including Figs. 3, 4, 5, and S1-S8, is available on Github, https://github.com/Crown1983/DNA-Equilizer-Gate and is available at Zenodo 10.5281/zenodo.4059727[43,44].

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

## Acknowledgements

The authors thank the Fundamental Research Funds for the Central Universities (No. YJ201975), the Natural Sciences and Engineering Research Council of Canada (No. RGPIN05240), and the Ontario Ministry of Research, Innovation and Science for financial supports. F.L. and C.F. thank the Ontario-China Young Scientist Exchange Program for the financial supports.

## Author contributions

G.A.W. and F.L. conceived idea and designed all experiments. G.A.W. performed all simulation experiments. G.A.W., X.X., H.M., F.C. performed all wet-lab experiments. G.M. and A.L.S. collected clinical samples and performed sample exaction and analysis. All authors contributed to the data analyses. G.A.W., F.C., and F.L. wrote the manuscript and all authors contributed to the revision of the manuscript.

## Competing interests

F.L. and G.A.W. are inventors of a patent application (PCT/CN2020/119612) that covers the design and uses of DNA Equalizer Gate technology for nucleic acid analyses. All other authors declare no competing interests.
