## [Peer Review File · Nature Communications]

Reviewers' Comments:

Reviewer #1:

Remarks to the Author:

Review for "Expanding detection windows for discriminating single nucleotide variants using rationally designed DNA equalizer probes" by Guan Wang, Feng Li, et al. In this work, the authors use the idea of shorter complementary oligos to disrupt reannealing of a double-stranded DNA, allowing effective discrimination of single nucleotide variants across a wider range of concentrations. The core idea is novel, and I would support eventual publication in Nature Communications.

That being said, there are some critical experiments that are missing to support generalizability, and a few limitations of the approach need to be highlighted. Furthermore, the paper is written and the figures are drawn in a needless complicated abstract way that inhibits, rather than facilitates, reader comprehension. I would be happy to review a revised version of the manuscript.

Major Comments:

1. The core idea is very nice, but the figures are currently drawn in a very abstract way that makes understanding the system needlessly difficult. Modules are named "Splitter", "Annihilation 1", "Annihilation 2" and "Signal Reporter". I understand that the authors are aiming to draw a parallel to electronic circuit design, but this is not very much point to abstraction when there's only one known implementation. (For comparison, there are many ways to implement a logical AND gate). The effect of this presentation decision is that casual molecular diagnostic readers would think that the current method is very far removed from reality, whereas I actually think it would potentially be adapted without too much effort. I would strongly recommend Figure 2 to be drawn in a more direct way with the DNA targets and probes present, a denaturation reaction, and a renaturation reaction. The current abstract illustration should go in the supplementary materials.

2. The authors must prominently discuss the potential for misinterpretation of concentration due to the non-monotonic nature of the response curve. The standard sigmoidal Signal-[Target] response curve is monotonic, meaning each Signal value maps to exactly 1 Target concentration. With the author's current implementation, there are two very different Target concentrations that generate the same Signal. This could result in ambiguity that would be extremely worrying for clinical applications. For example, if I detect an individual's HIV viral titer and the signal indicates EITHER 4 / mL (very low, HIV under control) or 4×10^5 / mL (very high, HIV drug resistance or patient non-compliance), then this does not inform a clear actionability. Without a reasonable way to address this point (e.g. through a separate non-DEG internal control), I just don't see how this can ever be FDA cleared and implemented in clinical assays.

3. The authors show experimental data for SNV discrimination for only 1 single SNV in Fig. 5 and Fig. 6. Supplementary Figure S18 shows 3 mutants, but all at the same position. This is deeply unsatisfying for a paper aimed to improving the robustness of detection, as it leaves the reader wondering whether the specific mutation/position detected is the only one out of many experimentally tested mutations that worked. For a paper of Nature Communications caliber, I would expect at least 12 different variant sequences to be tested (ideally closer to 100), with representative coverage of all 12 base transitions, as well as a range of different positions in the DEP probes, including some positions likely to fail (e.g. the first and last nucleotides of the DEG). This will give the readers a much better sense of what is possible and what is not possible with the paper's approach.

Minor Comments:

1. I think the signal reporter is not a core component of the innovation. There are many other ways to report double-stranded DNA but not single-stranded DNA. One way is to ligate adapters

on the end and perform NGS, which would provide massively increased multiplexing and much better sensitivity (following PCR amplification). Recommend to remove from Figure 2.

2. I would recommend the authors to call their 3-stranded waste complex a "3-stranded waste complex" rather than "triplex", because "triplex" DNA is a very specific structured formed through Hoogsteen binding, and is not an apt description of the waste product.

3. I would recommend for the authors to clearly indicate on their figures the nucleotide positions along the input X oligo where single nucleotide variants can be clearly discriminated (both theoretically, and experimentally tested).

4. The authors should not switch terminology from X+ to A and X- to B, etc. Adding to confusion, in Fig. 5, the X or AB species is referred to as Target. The constant switching of terminology is very annoying for readers.

Reviewer #2:

Remarks to the Author:

The manuscript titled "Expanding detection windows for discriminating single nucleotide variants using rationally designed DNA equalizer probes" by Wang et al. describes a strategy to improve the detection of target dsDNA, by suppressing the signal from spurious targets. The assay uses the principle of molecular sinks/reservoirs by applying a pair of probes (DNA equalizer probes, DEPs) to convert a double-stranded DNA (dsDNA) into single-stranded (ssDNA). This reviewer feels the approach is potentially interesting, but has the following concerns.

1) The use of DNA oligonucleotides as a sink to regenerate ssDNA strands has been explored for quite a while now in DNA circuits. The authors should compare their approach with other molecular sinks or reservoirs, instead of only with toehold switches or molecular beacons, to evaluate the novelty of the approach.

2) The authors claim that their approach can be used regardless of the identity or position of mutations. This feels as an overstatement. In fact, the approach is highly dependent on the sequence and concentration of the DEPs. Ideally, the pair of DEPs should cover the full length of the dsDNA targets to generate ssDNA. And this must be considered with respect to both the sequence and concentration of the different dsDNA targets (correct : spurious targets).

The authors should discuss these aspects and how such information could be acquired beforehand in real samples. Experimental evaluation of performance across different target conditions (sequence, length and concentration ratio) should also be performed.

3) The current model presented (Fig 3b) solves multiple reactions as if they are independent equations. This is inaccurate as the system is interconnected and the rate of reactions will change dynamically as different reactants are being used up.

4) The formation of AB product will preclude BCD product formation. As such, the uncorrected probability model (when $[X] \leq [DEP]$) is also inaccurate. The probability model used in case two should be applied across all $[X]$.

5) The authors simulated (Fig 4) and experimentally verified (Fig 5) that increasing the amount of DEPs can improve the upper range of target sensitivity with no effect on the lower range of sensitivity. This could be misleading and a discussion of trade-offs and optimization process on the concentration of DEPs should be included. For example, I would imagine that with increasing amount of DEPs used, the generated ssDNA target would have to compete against the unbound DEPs for the toehold probe, rendering the kinetics of the reaction less favorable, reducing the

absolute signal. This effect would be exacerbated with a low amount of targets.

6) The system is not very sensitive for measuring heterogeneous mutations. Using a cutoff of 0.35 FAM/Cy5 ratio (Fig S34-35), it appears that the SNP must be present at least 40-50% of the dsDNA population (Fig S36) in order to be detected. This makes the assay more limited as many diseases have SNP frequencies < 1%. It would be interesting to see if the authors can improve the system for better SNP sensitivity.

7) More clinical samples could be included, so that both the training (currently done with synthetic DNA standards) and validation sets could be done with clinical samples.

8) The ability to detect drug resistance in clinical samples has not been demonstrated, as there are no samples with positive drug resistance. Related claims should be modified to mention this as a potential application.

Response Letter

We thank both reviewers for their insightful comments and suggestions. We have thoroughly revised our manuscript accordingly. A point-by-point response is provided as following.

Reviewer #1:

Review for “Expanding detection windows for discriminating single nucleotide variants using rationally designed DNA equalizer probes” by Guan Wang, Feng Li, et al. In this work, the authors use the idea of shorter complementary oligos to disrupt reannealing of a double-stranded DNA, allowing effective discrimination of single nucleotide variants across a wider range of concentrations. The core idea is novel, and I would support eventual publication in Nature Communications.

That being said, there are some critical experiments that are missing to support generalizability, and a few limitations of the approach need to be highlighted. Furthermore, the paper is written and the figures are drawn in a needless complicated abstract way that inhibits, rather than facilitates, reader comprehension. I would be happy to review a revised version of the manuscript.

Major Comments:

Comment-1: The core idea is very nice, but the figures are currently drawn in a very abstract way that makes understanding the system needlessly difficult. Modules are named “Splitter”, “Annihilation 1”, “Annihilation 2” and “Signal Reporter”. I understand that the authors are aiming to draw a parallel to electronic circuit design, but this is not very much point to abstraction when there’s only one known implementation. (For comparison, there are many ways to implement a logical AND gate). The effect of this presentation decision is that casual molecular diagnostic readers would think that the current method is very far removed from reality, whereas I actually think it would potentially be adapted without too much effort. I would strongly recommend Figure 2 to be drawn in a more direct way with the DNA targets and probes present, a denaturation reaction, and a renaturation reaction. The current abstract illustration should go in the supplementary materials.

Response-1: We thank the reviewer for this suggestion, and we do agree that Figure 2 needs to be drawn in a more simple and straightforward manner to enhance the readership. Therefore, we have completely re-drawn the Figure 2 by 1) including an overall workflow for DEG as Figure 2a and 2) detailed denaturation and renaturation reactions in Figure 2b, 2c and 2d. We have also thoroughly revised the figure caption and results section to better reflect the content in revised Figure 2.

Revision-1: A new version of Figure 2 was added to the revised manuscript with revisions in the figure caption and discussion.

Revision in the manuscript:

Figure 2 | Schematic illustration of the DNA Equalizer Gate. (a) The overall workflow for quantifying dsDNA using DEG. A mixture of target dsDNA and DEPs is heated and rapidly cooled to produce ssDNA outputs with well-controlled quantity using an autonomous molecular computation in the test tube. Fluorescence signals are then generated *via* a reporter probe. (b) Mechanistically, the dsDNA target (AB) is denatured into A and B during a heating and snap cooling procedure. A competition between DEPs (C and D) and A then occurs for hybridizing with B during renaturation. The net amount of the ssDNA output (A) is quantitatively determined by an autonomous computing process that compares the initial concentrations between the target and DEPs. (c) When $[AB] \leq [DEPs]$, the reaction between B and DEPs (i.e. the formation of BCD) is thermodynamically favored, which maximizes the production of A . (d) When $[AB] > [DEPs]$, BC and BD are generated as intermediates, which then consumes A through strand displacement. (e) Through this computing process, DEG transforms the quantitative

relationship between the detection signal and target concentrations from a typical sigmoidal function to a unimodal function. As such, detection signals for a spurious target is significantly suppressed, enabling a much-enlarged detection window and improved discrimination factor (DF).

The workflow and principle of DEG are illustrated in Fig. 2. A double-stranded input **AB** produces a single-stranded target **A** and its complementary sequence **B** through a rapid heating at 95 °C and then snap cooling to 0 °C in a thermal protocol. **B** is then consumed by DEPs that are of the identical sequences with **A** forming three-stranded complex **BCD** (Fig 2b). The yield (η) of **A** is thus determined quantitatively by the concentration of DEPs. When the concentration of **AB** is less than those of DEPs, **A** is the predominant product, although a competition exists between **A** and DEPs for hybridizing to **B** (Fig. 2c). When the concentration of **AB** is greater than those of DEPs, unconsumed **B** will rehybridize with **A** in renaturation process (Fig. 2d). Therefore, a maximum yield exists when the concentration of **AB** equals to those of DEPs. Finally, the net **A** is quantified using a toehold-exchange reporter which is designed to be sensitive to SNVs. As each DEP is designed to only contain the sequence of either the toehold domain or the branch migration domain of the reporter, no fluorescence signal can be produced in the absence of the target (Fig. S15). Through DEG, a conventional sigmoid detection curve of hybridization probes is transformed into an asymmetric unimodal one (Fig. 2d).

Comment-2: The authors must prominently discuss the potential for misinterpretation of concentration due to the non-monotonic nature of the response curve. The standard sigmoidal Signal-[Target] response curve is monotonic, meaning each Signal value maps to exactly 1 Target concentration. With the author's current implementation, there are two very different Target concentrations that generate the same Signal. This could result in ambiguity that would be extremely worrying for clinical applications. For example, if I detect an individual's HIV viral titer and the signal indicates EITHER 4 / mL (very low, HIV under control) or $4 \cdot 10^5$ / mL (very high, HIV drug resistance or patient non-compliance), then this does not inform a clear actionability. Without a reasonable way to address this point (e.g. through a separate non DEG internal control), I just don't see how this can ever be FDA cleared and implemented in clinical assays.

Response-2: We thank the reviewer for raising this insightful concern. This is exactly the issue for the traditional hybridization probes and the reason why we introduce the DEG approach. For a traditional hybridization probe, there would be two standard sigmoidal signal-[Target] response curves (shown in Figure R1, left panel), one corresponds to non-resistant HIV infection and one corresponds to the drug-resistance infection. Therefore, the same detection signal could be a very low concentration of HIV under control or could be a very high concentration of HIV drug resistance, exactly as the reviewer worried. On the other hand, in our DEG approach, the same detection signal will only correspond to the non-resistant HIV infection, because the signal for the drug-resistant infection is significantly suppressed (Figure R1, right panel). There is indeed an issue for quantifying the exact target concentration (e.g. quantifying the viral load), as the same signal corresponds to two possible concentrations (both are non drug-resistance). In the revised manuscript, we demonstrated that a simple dilution strategy could effectively help narrow the concentration range, which was included in the revised manuscript as Fig. S37. We also

revised the discussion section of the manuscript to better reflect the feature of DEG and to address this comment.

Figure R1. Comparison between DEG and traditional approaches.

Revision-2: We have included a new set of experiments in the revised manuscript, which was summarized as Supporting Information Section S7 and Fig. S37. We have also revised the main content of the manuscript by adding the discussion of this scenario.

Revision in the manuscript:

Because of the mathematical transformation, the same detection signal may correspond to two concentrations of the same correct target (Fig. S37). This will not cause any issue for discriminating single nucleotide mutations, as the detection signal remains to be higher than any concentration of the mutated targets. For further quantifying the correct target using DEG, we found that the inclusion of a dilution step would effectively narrow the target concentration (Fig. S37).

Revision in Supplementary Material:

S7. Narrowing concentration ranges for quantifying dsDNA targets using DEG

Fig. S37 | Schematic illustration of DEG for discriminating single nucleotide mutation and quantifying unmutated targets. Once measured, the detection signals (red dots) can be used directly for quantitative analysis, as the detection signal is higher than that of the mutated target throughout the concentration range. However, for

quantifying a correct target, the same detection signal corresponds to two possible concentrations of the same target. This issue can be addressed by adding a simple dilution step. For example, we experimentally measured a detection signal of 0.35, which might be either 1 μM or 15 nM of the correct dsDNA target. However, after a $2 \times$ dilution step, the original 1 μM target concentration (500 nM after dilution) produced an even higher detection signal (0.8) whereas the signal of the original 15 nM target concentration (7.5 nM after dilution) decreased to 0.2. By doing so, we can easily narrow down the concentration of the target.

Comment-3: The authors show experimental data for SNV discrimination for only 1 single SNV in Fig. 5 and Fig. 6. Supplementary Figure S18 shows 3 mutants, but all at the same position. This is deeply unsatisfying for a paper aimed to improving the robustness of detection, as it leaves the reader wondering whether the specific mutation/position detected is the only one out of many experimentally tested mutations that worked. For a paper of Nature Communications caliber, I would expect at least 12 different variant sequences to be tested (ideally closer to 100), with representative coverage of all 12 base transitions, as well as a range of different positions in the DEP probes, including some positions likely to fail (e.g. the first and last nucleotides of the DEG). This will give the readers a much better sense of what is possible and what is not possible with the paper's approach.

Response-3: We fully agree with the reviewer that it is critical to evaluate the robustness and generalizability of the DEG approach using different sets of targets and different types/positions of mutations. Therefore, we have designed a series of new tests for 3 more sets of targets/mutations of varying length (Fig. S17-S19), 12 more sets of targets with 3 types of mutations at 4 representative positions (Fig. S25-S28), and 9 sets of clinically important single nucleotide variants frequently occurred in cancer (Fig. S33-S35). By including the new 36 target/mutation sequences in the revised manuscript, we believe that the readers will have a much better sense of the robustness of the DEG approach. We have also revised the Result section of the manuscript to better reflect the flexibility and limitation of the DEG approach.

Revision-3: We have performed the tests for targets of varying length (Fig. S17-S19), mutations of varying types and locations (Fig. S25-S28), and new sets of clinically important SNVs (Fig. S33-S35). We have also revised the manuscript by adding additional discussions of the new sets of targets and test results.

Revision in the manuscript:

The effectiveness and flexibility of DEG were further verified experimentally against varying length of dsDNA targets (Fig. S17-S19), varying types and locations of single nucleotide mutations (Fig. S25-S28), and finally 9 sets of clinically important SNVs (Fig. S33-S35). DEG works well for all sets of targets except when mutation occurs at the very edge of the dsDNA (Fig. S25).

Revision in Supplementary Material:

Revised by adding sections S4 and S5.3 and Fig. S17-S19, Fig. S25-S28, Fig. S33-S35.

S4. Length effect of target and DEP

Fig. S17 | Sequence for varying length of TT targets and corresponding DEPs to validate the length effect of target/DEPs.

Fig. S18 | Experimentally measured yields, DFs, and RFs for varying length of dsDNA targets from 87 bp to 32 bp using corresponding DEPs at concentrations of 200 nM, These results suggest that our DEG approach is workable for targets with varying length ranges with minimal impact to the analytical performance.

Fig. S19 | Experimentally measured yields, DFs, and RFs for target TT-32 using DEPs at concentration of 50, 100, and 200 nM, respectively.

Fig. S25 | Schematic illustration of analyzing varying single nucleotide mutations at different positions of TT-28 target.

Fig. S26 | Comparison of experimentally measured yield of correct target (green curve) and different single-nucleotide-mutations at varying positions (red curve) in target TT-28. Concentration of DEPs were fixed at 200 nM.

Fig. S27 | Experimentally measured DF of target TT-28 with different mutations.

Fig. S28 | Experimentally measured RF of target TT-28 with different mutations.

S5.3 Detection of clinically important single nucleotide variants in cancer

To further demonstrate the versatility and robustness of our DEG method, we designed nine sets of DEG and toehold-exchange probes for clinically important single nucleotide variants frequently detected in cancer. The sequences and designs are shown in Fig. S33 and the performance of DEG for analyzing the 9 sets of targets are shown in Fig. S34 and S35.

Fig. S33 | Designs and sequences for nine sets of clinically important single nucleotide variants frequently encountered in cancer.

Fig. S34 | Experimentally measured yields, DFs, and RFs for analyzing BRAF-D594G, BRAF-V600E, EGFR-G7119A, EGFR-L858R and EGFR-L861Q. The concentration of DEG is fixed at 200 nM.

Fig. S35 | Experimentally measured yields, DFs, and RFs for analyzing KRAS-G12A, KRAS-G13V, PIK3CA-H1047R and STK11-F354L. The concentrations of DEPs were fixed at 200 nM.

Minor Comments:

Comment-4: I think the signal reporter is not a core component of the innovation. There are many other ways to report double-stranded DNA but not single-stranded DNA. One way is to ligate adapters on the end and perform NGS, which would provide massively

increased multiplexing and much better sensitivity (following PCR amplification). Recommend to remove from Figure 2.

Response-4: The signal reporting component was removed in the revised manuscript.

Comment-5: I would recommend the authors to call their 3-stranded waste complex a “3-stranded waste complex” rather than “triplex”, because “triplex” DNA is a very specific structured formed through Hoogstein binding, and is not an apt description of the waste product.

Response-5: Agree! The terminology has been corrected in the revised manuscript.

Comment-6: I would recommend for the authors to clearly indicate on their figures the nucleotide positions along the input X oligo where single nucleotide variants can be clearly discriminated (both theoretically, and experimentally tested).

Response-6: We have included tests for mutations at 5 representative positions. The results suggest that DEG approach works for most of the mutation positions, except for the ones at the very end of the target. We have included this discussion in the revised manuscript.

Revision-6: We have revised the manuscript by discussing the limitation of DEG for discriminating mutations at the very end of the target.

In the manuscript:

DEG works well for all sets of targets except when mutation occurs at the very edge of the dsDNA (Fig. S25).

Comment-7: The authors should not switch terminology from X+ to A and X- to B, etc. Adding to confusion, in Fig. 5, the X or AB species is referred to as Target. The constant switching of terminology is very annoying for readers.

Response-7: Agree! All X species have been revised to AB in the revised manuscript.

Reviewer #2:

The manuscript titled “Expanding detection windows for discriminating single nucleotide variants using rationally designed DNA equalizer probes” by Wang et al. describes a strategy to improve the detection of target dsDNA, by suppressing the signal from spurious targets. The assay uses the principle of molecular sinks/reservoirs by applying a pair of probes (DNA equalizer probes, DEPs) to convert a double-stranded DNA (dsDNA) into single-stranded (ssDNA). This reviewer feels the approach is potentially interesting, but has the following concerns.

Comment-1: The use of DNA oligonucleotides as a sink to regenerate ssDNA strands has been explored for quite a while now in DNA circuits. The authors should compare their approach with other molecular sinks or reservoirs, instead of only with toehold switches or molecular beacons, to evaluate the novelty of the approach.

Response-1: We thank the reviewer for this suggestion. To the best of our knowledge, there are two types of molecular Sinks or Reservoirs commonly used in DNA circuits and DNA nanotechnology. The first type of Sink module often serves as a threshold to “drain” the templet or target strands (e.g., Fig. R2a, Montagne, et. al. Mol. Syst. Biol. 2010, 7, 466; Fig. E2b, Montagne, et al. Nat. Commun. 2016, 7, 13474). This type of molecular sink has been used to help clean the background for nucleic acid amplification or to design oscillation DNA circuits, which is not quite relevant to our work. The second type of molecular sink often used in designing ultraspecific strategies for discriminating single nucleotide variants (e.g., Zhang, et. al. Nat. Chem. 2015, 7, 545-553; Seelig, et. al. JACS 2016, 138, 5076-5086) is quite relevant to our approach and we have thus promptly compared DEG with strategies using this type of molecular Sink or Reservoir.

The field of DNA nanotechnology has also offered several strategies for analyzing dsDNA targets by generating ssDNA or ssDNA domains. For example, Zhang et al. introduced X-probe for analyzing dsDNA by attaching toehold domains to dsDNA target through asymmetric PCR and annealing (Fig. R3a, Zhang, et al. Nat. Chem. 2013, 5, 782-789). Ellis et al. introduced toehold domains to PCR amplicons by using uracil glycosylase (Fig. R3b, Ellis et al. JACS 2013, 135, 5612-5619). Komiyama, et al. converted dsDNA target into ssDNA by using Exonuclease and PNA (Fig. R3c, Komiyama et al. Nucleic Acid Res. 2004, 32, e42). Comparing to existing strategies, our DEG approach offers two advantages: 1) produce detectable ssDNA without the need for enzymes and tedious procedures; and 2) quantitatively suppress the detection signals for the mutated targets, which have never been demonstrated before. Therefore, we revised our manuscript by comparing DEG with existing strategies for discriminating single nucleotide mutations in dsDNA.

Figure R3. Current strategies for analyzing dsDNA targets using DNA nanotechnology.

Revision-1: The manuscript has been revised by adding comparison between DEG approach with existing strategies for analyzing mutations in dsDNA and corresponding references.

In the manuscript:

This differs significantly with existing strategies where enzymes and tedious procedures were often used to generate toehold domains for dsDNA^{17,43} or to remove antisense strands to produce ssDNA⁴⁴.

Unlike existing molecular sink or reservoir created to enhance sequence selectivity,^{36,38} the design of which requires prior knowledge of the specified mutation, the DEG splits a dsDNA and thus produces a corresponding sink for any mutation.

Comment-2: The authors claim that their approach can be used regardless of the identity or position of mutations. This feels as an overstatement. In fact, the approach is highly dependent on the sequence and concentration of the DEPs. Ideally, the pair of DEPs should cover the full length of the dsDNA targets to generate ssDNA. And this must be considered with respect to both the sequence and concentration of the different dsDNA targets (correct : spurious targets). The authors should discuss these aspects and how such information could be acquired beforehand in real samples. Experimental evaluation of performance across different target conditions (sequence, length and concentration ratio) should also be performed.

Response-2: We thank the reviewer for raising this critical concern and we agree that with the original sets of tests, it is difficult to demonstrate the robustness and versatility of the DEG approach. To better evaluate the robustness of DEG, we performed a series of new tests, including 3 more sets of targets/mutations of varying length (Fig. S17-S19), 12 more sets of targets with 3 types of mutations at 4 representative positions (Fig. S25-S28), and 9 sets of clinically important single nucleotide variants frequently occurred in cancer (Fig.

S33-S35). We have also revised the Result section of the manuscript to better reflect the flexibility and limitation of the DEG approach.

The reviewer also made a very good point that the DEG approach is highly dependent on the sequence and concentration of DEPs and this is exactly why we systematically studied the sequence design and concentration effect of DEPs both *in silico* and experimentally. Practically, both sequence and concentration of DEPs can be designed and controlled by the end user as we demonstrated in Figures 4 (*in silico*) and 5 (experimentally). The sequence information of the target is readily available using bioinformatic tools such as BLAST and the length of the dsDNA target can also be controlled by designing appropriate primers as we demonstrated in Figure 6 and 7.

Revision-2: We have performed the tests for targets of varying length (Fig. S17-S19), mutations of varying types and locations (Fig. S25-S28), and new sets of clinically important SNVs (Fig. S33-S35). We have also revised the manuscript by adding additional discussions of the new sets of targets and test results.

Revision in the manuscript:

The effectiveness and flexibility of DEG were further verified experimentally against varying length of dsDNA targets (Fig. S17-S19), varying types and locations of single nucleotide mutations (Fig. S25-S28), and finally 9 sets of clinically important SNVs (Fig. S33-S35). DEG works well for all sets of targets except when mutation occurs at the very edge of the dsDNA (Fig. S25).

Revision in Supplementary Material:

Revised by adding sections S4 and S5.3 and Fig. S17-S19, Fig. S25-S28, Fig. S33-S35.

Comment-3: The current model presented (Fig 3b) solves multiple reactions as if they are independent equations. This is inaccurate as the system is interconnected and the rate of reactions will change dynamically as different reactants are being used up.

Response-3: We apologize for the confusion made in this part. As shown in Fig. 3a, our model has taken all 8 element reactions at their equilibrium states (validated by the stability experiment in Fig. S14) into consideration, so it does reflect the global thermodynamics of the DEG process. As the reviewer suggested, all reactions in the system are interconnected, so mathematically some of the equations are coupled and thus are difficult to solve. So, the introduction of reaction matrix and ranking independent equations in Fig. 3b is purely a mathematical operation to help find meaningful answers for this highly complexed reaction network. To better explain the model building and mathematical operation, we have revised the result part of the manuscript and added detailed discussion of the matrix ranking in the supporting information.

Revision-3: Revision has been made to the manuscript to better explain the model building and mathematical calculation. Detailed discussion of the matrix and ranking have also been added to the supporting information.

Revision in the manuscript:

Theoretical model. Here, a mathematical model was introduced to quantitatively profile DEG by taking all possible reactions into consideration (Fig. 3a). To derive the yield of each DNA species in this reaction network as a function of both sequence design ($\Delta\Delta G^0$)

and equalizer probe concentrations, a set of eight equilibrium equations need to be solved. However, we found that these equations were coupled to one another, which was mathematically difficult to solve. Therefore, a stoichiometric matrix RM was introduced to help simplify the calculation (Fig. 3a), where the first four rows were ranked to be essential (details in Supplementary Information section S2.4). This essential set of equilibrium equations was then solved by a numerical approach, where distributions of A and AB were solved as a function of the target concentration and plotted in Fig. 3c.

Revision in Supplementary Material:

S2.4 DNA Equalizer Gate

DEG is designed to convert a dsDNA target into a ssDNA output in a quantitative manner with well-defined detection window. To simulate this process, we consider that all reactions are thermodynamically driven, and all DNA species are in their thermodynamic stable states. Under this assumption, a set of equilibrium equations could be used to predict the concentration distribution of newly formed DNA species (Fig. 3a in the main content). However, only independent equations need to be solved otherwise meaningless answers will be generated. To help determine independent equilibrium equations, we extract a numerical reaction matrix (RM) from the reaction system:

$$RM = \begin{bmatrix} -1 & -1 & 1 & 0 & 0 & 0 & 0 & 0 \\ 0 & -1 & 0 & -1 & -1 & 1 & 0 & 0 \\ 0 & -1 & 0 & -1 & 0 & 0 & 1 & 0 \\ 0 & -1 & 0 & 0 & -1 & 0 & 0 & 1 \\ 0 & 0 & 0 & 0 & -1 & 1 & -1 & 0 \\ 0 & 0 & 0 & -1 & 0 & 1 & 0 & -1 \\ -1 & 0 & 1 & 1 & 0 & 0 & -1 & 0 \\ -1 & 0 & 1 & 0 & 1 & 0 & 0 & -1 \end{bmatrix};$$

In RM, each row represents a possible chemical reaction and Reactant is the column of all DNA species. The rank of RM is 4 (validated by Matlab), which is less than the dimension of RM. As such, only 4 independent equations are existing in this reaction system and we choose the first four reactions in our model. All ΔG_{rxn}^0 values are predicted using NUPACK and the equilibrium equations are shown below:

$$\frac{[AB]_{eq}}{[A]_{eq} \cdot [B]_{eq}} = K_{eq,i} = e^{-\Delta G_i^0/RT}$$

$$\frac{[BCD]_{eq}}{[B]_{eq} \cdot [C]_{eq} \cdot [D]_{eq}} = K_{eq,ii} = e^{-\Delta G_{ii}^0/RT}$$

$$\frac{[BC]_{eq}}{[B]_{eq} \cdot [C]_{eq}} = K_{eq,iii} = e^{-\Delta G_{iii}^0/RT}$$

$$\frac{[BD]_{eq}}{[B]_{eq} \cdot [D]_{eq}} = K_{eq,iv} = e^{-\Delta G_{iv}^0/RT}$$

where $[A]_0 = [A]_{eq} + [AB]_{eq}$; $[B]_0 = [B]_{eq} + [BCD]_{eq} + [BC]_{eq} + [BD]_{eq}$;

$[C]_0 = [C]_{eq} + [BC]_{eq}$; $[D]_0 = [D]_{eq} + [BD]_{eq}$;

$[A]_0, [B]_0, [C]_0,$ and $[D]_0$ are initial concentrations;

Standard reaction free energies are calculated at 4 °C according to the experimental condition of DEG.

Comment-4: The formation of AB product will preclude BCD product formation. As such, the uncorrected probability model (when $[X] \leq [DEP]$) is also inaccurate. The probability model used in case two should be applied across all $[X]$.

Response-4: We are afraid that we cannot agree with the reviewer for this comment. When $[X] \leq [DEP]$, the thermodynamic model has already taken the competing process between the formation of AB and BCD into consideration, so it is not necessary to correct it with the probability model. As an evidence, our simulation and experimental observation agrees well with each other at this concentration range. We tried to further apply probability correction to this range (as suggested by the reviewer), and this would lead to a final yield greater than 100%, which is mathematically invalid. When $[X] < [DEP]$, the probability correction is mainly made for the competitive formation between BCD and BC or BD, rather than the formation of AB. We hope this explanation would help clear the confusion for the reviewer.

Comment-5: The authors simulated (Fig 4) and experimentally verified (Fig 5) that increasing the amount of DEPs can improve the upper range of target sensitivity with no effect on the lower range of sensitivity. This could be misleading and a discussion of trade-offs and optimization process on the concentration of DEPs should be included. For example, I would imagine that with increasing amount of DEPs used, the generated ssDNA target would have to compete against the unbound DEPs for the toehold probe, rendering the kinetics of the reaction less favorable, reducing the absolute signal. This effect would be exacerbated with a low amount of targets.

Response-5: We thank the reviewer for raising this concern and indeed we had the very same concern when developing DEG. For the optimization on the concentration of DEPs, we have studied this effect in a quantitative manner both in silico and experimentally, as shown in Fig. 5. Our model is established based on thermodynamics, so it did not take possible cross-reaction between DEP and toehold-exchange beacons into consideration. Fortunately, as we demonstrated in Figure S15, no apparent cross-reaction was found between DEP and the toehold-exchange probe even when DEPs were at micromolar concentrations. To better illustrate this point, we revised Figure S15 to better reflect the cross-reactivity between DEPs and the toehold probe. We also highlighted the concentration ranges, where DEP concentrations were used throughout this work.

Revision-5: Revision has been made to Figure S15 and its caption.

Fig. S15 | Estimation of signal leakage from DEPs. A possible source of fluorescence background is the signal caused by the interaction of DEPs and the reporter probe. Therefore, we estimated the signal leakage at varying DEP concentrations. **a.** Schematic

illustration of the signal leakage caused by the interaction among DEPs and the reporter. **b.** Estimated leakage (w/o Target) as a function of DEP concentrations, which is also compared to the target-specific fluorescence (w/ Target). The target concentration is fixed at 10 nM, concentrations of DEPs are varied from 10 nM to 5 μ M. No apparent fluorescence signal was observed when treating the fluorescence probe with up to 1 μ M DEPs, suggesting that there was no cross-reaction between DEPs and the probe and thus there was no competition between DEPs and ssDNA output for the probe. Each error bar represents one standard deviation from duplicate analyses.

Comment-6: The system is not very sensitive for measuring heterogeneous mutations. Using a cutoff of 0.35 FAM/Cy5 ratio (Fig S34-35), it appears that the SNP must be present at least 40-50% of the dsDNA population (Fig S36) in order to be detected. This makes the assay more limited as many diseases have SNP frequencies < 1%. It would be interesting to see if the authors can improve the system for better SNP sensitivity.

Response-6: Figures S34 and S35 in the original manuscript was to determine the detection limit of DEG rather than to determine the low fraction (frequency) mutations. The value 0.35 is the ratio between two fluorescent channels rather than mutation abundance. We apologize for the confusion and have revised these figures to better reflect the detection limit of the DEG approach (Fig. S46 and S47).

As shown in Fig S36a in the original manuscript (Fig. 48 in the revised manuscript), we were able to get dateable signal at a mutation fraction of 10%. To better push the detection of low fraction mutations in the presence of high abundant wild-types, we performed a new set of experiment as shown in Fig. S36, as low as 0.5% mutation fraction (frequency) could be clearly discriminated.

Revision-6: Revisions have been made to Fig. S46 and S47. A new section has been added to the Sup. Info. as Section 6 and a new set of results were included as Fig. S36.

S6. Evaluation of DEG for detection of rare mutations

Fig. S36 | As low as 0.5% mutated targets in the background of high concentrations of unmutated sequences can be detected effectively using DEG.

Fig. S46 | The detection limit of the 4-DEP, dual reporter DEG for analyzing synthetic DNA targets either drug resistant mutant that is drug resistant positive (D.R.+) or a wild-type that is drug resistant negative (D.R.-). a. Normalized fluorescence in both FAM and Cy5 channels as a function of target concentrations for the detection of drug resistant positive mutant. b. Normalized fluorescence in both FAM and Cy5 channels for the detection of a wild-type target. c. The dual-channel fluorescence distribution map for targets (either D.R.+ or D.R.-) spanning over 0.16, 0.31, 0.62, 1.25, 2.5, 5, 10, 20, 40, and 80 nM. The detection limit was found to be 0.62 nM for drug resistant positive target while 1.25 nM for drug resistant negative target. Gray shading area in the subplot indicating the fluorescence distribution that cannot distinguish D.R.+ and D.R. - targets.

Fig. S47 | The detection limit of the 4-DEP, dual reporter DEG for analyzing synthetic DNA targets with 800 nM DEPs. a. Normalized fluorescence in both FAM and Cy5

channels as a function of target concentrations for the detection of drug resistant positive mutant. **b.** The fluorescence distribution of FAM and Cy5 channels as a function of mutant-type target concentration. **c.** The dual-channel fluorescence distribution map for targets (either D.R.+ or D.R.–) spanning over 0.16, 0.31, 0.62, 1.25, 2.5, 5, 10, 20, 40, and 80 nM. The detection limit was found to be 0.62 nM for both drug resistant positive and drug resistant negative targets. Gray shading area in the subplot indicating the fluorescence distribution that cannot distinguish D.R+ and D.R. – targets.

Comment-7: More clinical samples could be included, so that both the training (currently done with synthetic DNA standards) and validation sets could be done with clinical samples.

Response-7: We thank the reviewer for this suggestion. We have included 13 new clinical parasitic TT worm samples with no drug-resistant mutation into the training sets. Figure 7c was revised accordingly by including the new sets of clinical samples.

Revision-7: Revision made to Figure 7 and caption.

Figure 7 | Application of DEG-PCR to analyzing clinical parasitic worm samples. (a) A typical workflow for analyzing parasitic worm (*Trichuris trichiura*, TT) specimens collected from stool samples of school-age children in the rural areas of Honduras followed by the detection using DEG-PCR. **(b)** Simultaneous detection of parasitic infection and screening for drug-resistance was achieved using a dual-channel design (FAM- and Cy5-

Reporter). PCR primers were designed to amplify nucleotide 1246-1333 in the β -tubulin gene, containing the 200th codon. A single nucleotide A to T mutation of this codon is a hotspot for drug resistance screening. A toehold-exchange reporter (FAM-reporter, green duplex) labeled with FAM was used to discriminate this point mutation, whereas a strand-displacement reporter with no reverse toehold (Cy5-reporter, red duplex) was employed to detect a conservative region near codon 200. Experimental tests of the dual channel DEG-PCR using synthetic DNA standards (blue and red dots) and 13 (D.R.–) clinical samples (green circles) as a training set (**c**) and 8 unknown clinical parasitic worm samples (**d**). Test results are classified into three areas defining the positive infection and drug resistance (D.R.+), positive infection and no drug resistance (D.R.–), and negative infection (N.C.). Error eclipses with 99% confidence were used to define D.R.+ and D.R.–. Eight clinical worm specimens including six *Trichuris trichiura* worms (TT-1 to TT-6) and two *A. lumbricoides* worms (AL, as negative controls) were tested and plotted in **d**.

Comment-8: The ability to detect drug resistance in clinical samples has not been demonstrated, as there are no samples with positive drug resistance. Related claims should be modified to mention this as a potential application.

Response-8: Unfortunately (fortunate for the kids in Honduras), of all clinical samples we collected, we did not find any parasitic worm samples with drug resistance mutations. Therefore, we modified the claim to be a potential application. We have revised "...identification of drug resistance" into "**screen of drug resistance**" to better reflect the results.

Reviewers' Comments:

Reviewer #1:

Remarks to the Author:

Review for resubmission of "Expanding detection windows for discriminating single nucleotide variants using rationally designed DNA equalizer probes" by Guan Wang, Feng Li, et al. The authors have largely addressed my concerns from the initial draft, and I commend them for doing a thorough job on the revision. I believe that the revised manuscript is now close to its final form for publication in Nature Communications. Below are some additional recommendations that the authors may want to consider.

1. I am generally impressed by the results from the expanded experiments on different mutations, as well as the authors' experimental demonstration to use dilution to discrimination between the high and low concentration states. I think the authors should consider putting a summary of these new results from Fig. S26 and particularly Fig. S37 into the main text. If figure count/space is a limitation, I would suggest to merge Figs. 3 and 4 to make room. On a similar note, the authors should not feel embarrassed about the low performance of the mutations on the first base, as this is a fairly standard limitation of all DNA probes, including both Taqman probes and toehold probes.

2. In Fig. S33 and S34, EGFR-G7119A is presumably a typo of EGFR-G719A. Please check all other variant labels carefully for accuracy.

Reviewer #2:

Remarks to the Author:

I appreciate the authors' comprehensive effort in addressing my previous concerns. The manuscript has been greatly improved with the new figures and data.

Response Letter

We thank both reviewers for their newer version of comments. We have made further revisions accordingly. A point-by-point response letter is provided as following:

Reviewer #1:

Review for resubmission of “Expanding detection windows for discriminating single nucleotide variants using rationally designed DNA equalizer probes” by Guan Wang, Feng Li, et al. The authors have largely addressed my concerns from the initial draft, and I commend them for doing a thorough job on the revision. I believe that the revised manuscript is now close to its final form for publication in Nature Communications. Below are some additional recommendations that the authors may want to consider.

Comment 1: I am generally impressed by the results from the expanded experiments on different mutations, as well as the authors’ experimental demonstration to use dilution to discrimination between the high and low concentration states. I think the authors should consider putting a summary of these new results from Fig. S26 and particularly Fig. S37 into the main text. If figure count/space is a limitation, I would suggest to merge Figs. 3 and 4 to make room. On a similar note, the authors should not feel embarrassed about the low performance of the mutations on the first base, as this is a fairly standard limitation of all DNA probes, including both Taqman probes and toehold probes.

Response-1: We thank the reviewer for the suggestions. We have put results from Fig. S26 in the earlier version as Fig. 5d and 5e and moved results from Fig. S37 to Fig. 8 in the revised manuscript. Figure legends were also revised accordingly.

Comment-2: In Fig. S33 and S34, EGFR-G7119A is presumably a typo of EGFR-G719A. Please check all other variant labels carefully for accuracy.

Response-2: The typo has been corrected in the revised manuscript.

Reviewer #2:

I appreciate the authors' comprehensive effort in addressing my previous concerns. The manuscript has been greatly improved with the new figures and data.